# Unsupervised active pre-training for reinforcement learning

## ABSTRACT

We introduce a new unsupervised pre-training method for reinforcement learning called **APT**, which stands for **A**ctive**P**re-**T**raining. APT learns a representation and a policy initialization by actively searching for novel states in reward-free environments. We use the contrastive learning framework for learning the representation from collected transitions. The key novel idea is to collect data during pre-training by maximizing a particle based entropy computed in the learned latent representation space. By doing particle based entropy maximization, we alleviate the need for challenging density modeling and are thus able to scale our approach to image observations. APT successfully learns meaningful representations as well as policy initializations without using any reward. We empirically evaluate APT on the Atari game suite and DMControl suite by exposing task-specific reward to agent after a long unsupervised pre-training phase. On Atari games, APT achieves human-level performance on 12 games and obtains highly competitive performance compared to canonical fully supervised RL algorithms. On DMControl suite, APT beats all baselines in terms of asymptotic performance and data efficiency and dramatically improves performance on tasks that are extremely difficult for training from scratch. Importantly, the pre-trained models can be fine-tuned to solve different tasks as long as the environment does not change. Finally, we also pre-train multi-environment encoders on data from multiple environments and show generalization to a broad set of RL tasks.

## 1 INTRODUCTION

Deep reinforcement learning (RL) provides a general framework for solving challenging sequential decision-making problems, it has achieved remarkable success in advancing the frontier of AI technologies thanks to scalable and efficient learning algorithms (Mnih et al., 2015; Lillicrap et al., 2015; Schulman et al., 2015; 2017). These landmarks include outperforming humans in board (Silver et al., 2016; 2018; Schrittwieser et al., 2019) and computer games (Mnih et al., 2015; Berner et al., 2019; Schrittwieser et al., 2019; Vinyals et al., 2019; Badia et al., 2020a), and solving complex robotic control tasks (Andrychowicz et al., 2017; Akkaya et al., 2019). Despite these successes, a key challenge with Deep RL is that it requires a huge amount of interactions with the environment before it learns effective policies, and needs to do so for each encountered task. Environments are required to have carefully designed task-specific reward functions to guide the RL algorithms (Andrychowicz et al., 2017; Ng et al., 1999), which further limits its wide applications of Deep RL. This is in contrast to how intelligent creatures learn in the absence of external supervisory signals, acquiring abilities in a task-agnostic manner by exploring the environment.

Unsupervised pre-training is a framework that trains models without expert supervision has obtained promising results in computer vision (Oord et al., 2018; He et al., 2019; Chen et al., 2020b; Caron et al., 2020; Grill et al., 2020) and natural language modeling (Vaswani et al., 2017; Devlin et al., 2018; Peters et al., 2018; Brown et al., 2020). The key insight of unsupervised pre-training techniques is learning a good representation or initialization from a massive amount of unlabeled data such as ImageNet (Deng et al., 2009), Instagram image set (He et al., 2019), Wikipedia, and WebText (Radford et al., 2019) which are easier to collect and scales to millions or trillions of data points. As a result, The learned representation when fine-tuned on the downstream tasks can solve them efficiently without needing any supervision or in a few-shot manner.

Driven by the significance of the massive abundance of unlabeled data relative to labeled data, we pose the following question: *is enabling efficient unsupervised pre-training for deep RL as easy as increasing the amount of unlabeled data?* Unlike the computer vision or language domains, in reinforcement learning it's not obvious where to extract large pools of unlabeled data. A natural choice is pre-training on ImageNet and transfer the encoder to reinforcement learning tasks. We experimented with using ImageNet data for unsupervised representation learning as initialization of the encoder in deep RL agent, specifically, we used the momentum contrast (He et al., 2019; Chen et al., 2020c) method which is one of the state-of-the-art methods for representation learning. We

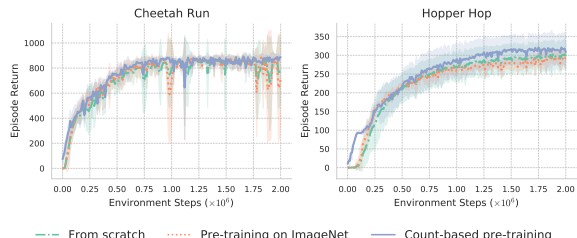

Figure 1: Unsupervised pre-training for deep RL on DM-Control. After pre-training (*e.g.* on ImageNet or in Cheetah reward free environment), the agent fine-tunes the pre-trained representation or initialization to achieve higher task-specific rewards (*e.g.* let the Cheetah run faster). ImageNet pre-training denotes training MoCo on downsampled ImageNet. Count-based pre-training means training RL agent with only count-based exploration signal. The training details are in Appendix Section F.1. The results show none of the two methods outperforms training from scratch.

used DrQ (Kostrikov et al., 2020) as the RL optimization algorithm. The results on DMControl are shown in Figure 1. We can see that using ImageNet pre-trained representations does not lead to any significant improvement over training from scratch. We also experimented with using supervised pre-trained ResNet features as initialization similar to Levine et al. (2016) (details in Appendix) but the results are no different. This seems in contrast to the preeminent successes of ImageNet pre-trained models in various computer vision downstream tasks (see *e.g.* Krizhevsky et al., 2012; Zeiler & Fergus, 2014; Hendrycks et al., 2019; Chen et al., 2020a). On the other hand, previous research in robotics also found that ImageNet pre-training did not help (Julian et al., 2020). We hypothesize that the reason for the discrepancy is that the ImageNet data distribution is far from the induced sample distribution encountered during RL training. It is therefore necessary to collect data from the RL agent induced distribution.

To investigate this hypothesis, we also experimented with training RL agents by 'exhaustively' collecting data during the reward-free interaction. Specifically, during pre-training phase, the only reward signal is defined by the count-based exploration (Bellemare et al., 2016; Ostrovski et al., 2017) which is one of the state-of-the-art methods for exploration (Taïga et al., 2019), and the density estimation model is PixelCNN (Van den Oord et al., 2016). The results of using the resulting pre-trained policy as initialization are shown in Figure 1. We can see that pre-trained initialization in Cheetah environment does not improve significantly over random initialization on Cheetah Run task. Similarly, pre-trained initialization in Hopper environment only leads to a small improvement over baseline. The reason for this ineffectiveness is that density modeling at the pixel level is difficult especially in the low-data and non-stationary regime. The results on DMControl demonstrate that simply increasing the amount of unlabeled data does not work well, therefore we need a more systematical strategy that caters to RL.

In this paper, we address the issue by proposing to actively collect novel data by exploring unknown areas in the task agnostic environment. Our means is maximizing the entropy of visited state distribution subject to some prior constraints. The entropy maximization principle (Jaynes, 1957) originated in statistical mechanics, where Jaynes showed that entropy in statistical mechanics and information theory were equivalent. Our motivation is that the resulting representation and initialization will encode both prior information while being as agnostic as possible, and can be adapted to various downstream tasks. While the entropy maximization principle seems simple, it is practically difficult to calculate the Shannon entropy (Shannon, 2001) as a density model is needed. To remedy this, we resort to the particle-based entropy estimator (Singh et al., 2003; Beirlant, 1997) which has wide applications in various machine learning areas (Sricharan et al., 2013; Pál et al., 2010; Jiao et al., 2018). The particle-based entropy estimator is known to be asymptotically unbiased and consistent. Specifically, it computes the average of the Euclidean distance of each sample to its nearest neighbors. We compute the entropy in the latent representation space, for this we adapt the idea of contrastive learning (Hadsell et al., 2006; Gutmann & Hyvärinen, 2010; Mnih & Kavukcuoglu, 2013; He et al., 2019; Chen et al., 2020b) to encode image observations to representation space.

Our approach alternates between training the encoder via contrastive learning and RL style optimization of maximizing expected reward where reward is defined by the particle-based entropy. After the pre-training phase, we can either fine-tune the encoder representation for test tasks that have different action space dimension or fine-tune the policy initialization for tasks with the same action space dimension. Since our method actively collects data during the pre-training phase, the method is named as **A**ctive **P**re-**T**raining (**APT**). We empirically evaluate APT on the Atari game suite and DMControl suite by exposing task-specific reward to the agent after a long unsupervised pre-training phase. On the full suite of Atari games, fine-tuning APT pre-trained models achieves human-level performance on 12 games. On the Atari $100k$ benchmark (Kaiser et al., 2019), our fine-tuning achieves $1.3\times$ higher human median scores than state-of-the-art training from scratch and $4\times$ higher scores than state-of-the-art pre-training RL. On DMControl suite, fine-tuning APT pre-trained models beating all baselines in terms of asymptotic performance and data efficiency and solving tasks that are extremely difficult for training from scratch.

The contributions of our paper can be summarized as: (i) We propose a new approach for pre-training in RL. (ii) We show that our pre-training method significantly improves data efficiency of solving downstream tasks on DMControl and Atari suite. (iii) We demonstrate that pre-training with particle-based entropy maximization in contrastive representation space significantly outperforms prior count-based approaches that rely on density modeling.

## 2 RELATED WORK

### 2.1 INTRINSIC MOTIVATION AND EXPLORATION

The learning process of RL agents becomes highly inefficient in sparse supervision tasks when relying on standard exploration techniques. This issue can be alleviated by introducing intrinsic motivation, *i.e.*, denser reward signals that can be automatically computed. These rewards are generally task-agnostic and might come from state visitation count bonus (Bellemare et al., 2016; Tang et al., 2017; Ostrovski et al., 2017; Zhao & Tresp, 2019), learning to predict environment dynamics (Meyer & Wilson, 1991; Pathak et al., 2017; Burda et al., 2018a; Sekar et al., 2020), distilling random neural networks (Burda et al., 2018b; Choi et al., 2018), hindsight relabeling (Andrychowicz et al., 2017), learning options (Sutton et al., 1999) through mutual information (Jung et al., 2011; Mohamed & Rezende, 2015), information gain (Lindley, 1956; Sun et al., 2011; Houthooft et al., 2016), successor features (Kulkarni et al., 2016; Machado et al., 2018), maximizing mutual information between behaviors and some aspect of the corresponding trajectory (Gregor et al., 2016; Florensa et al., 2017; Warde-Farley et al., 2018; Hausman et al., 2018; Shyam et al., 2019), using imitation learning to return to the furthest discovered states (Ecoffet et al., 2019), self-play curriculum (Schmidhuber, 2013; Sukhbaatar et al., 2017; Liu et al., 2019), exploration in latent space (Vezzani et al., 2019), injecting noise in parameter space (Fortunato et al., 2017; Plappert et al., 2017), learning to imitate self (Oh et al., 2018), predicting improvement measure (Schmidhuber, 1991; Oudeyer et al., 2007; Lopes et al., 2012; Achiam & Sastry, 2017), and unsupervised auxiliary task (Jaderberg et al., 2016). The work by Badia et al. (2020b) also considers k-nearest neighbor based intrinsic reward to incentive exploration, and shows improved exploration in sparse reward games. Our work differs in that we consider reward-free settings and the objective of our intrinsic reward is based on particle-based entropy instead of count bonus. The work closest to ours is Hazan et al. (2019) which presents provably efficient exploration algorithms under certain conditions. However, their method directly estimates state visitations through a density model which is difficult to scale. In contrast, our work turns to particle based entropy maximization in contrastive representation space. Concurrent work by Mutti et al. (2020) shows maximizing particle-based entropy can improve data efficiency in solving downstream continuous control tasks. However, their method relies on importance sampling and on-policy RL which suffers from high variance and is difficult to scale. In contrast, our work resorts to a biased but lower variance entropy estimator which is scalable for high dimensional observations and suitable for off-policy RL optimization.

### 2.2 DATA EFFICIENCY IN RL

Deep RL algorithms are sample inefficient compared to intelligent biological creatures, which can quickly learn to complete new tasks. To close this data efficiency gap, various methods have been proposed: Kaiser et al. (2019) introduce a model-based agent (SimPLe) and show that it compares

favorably to standard RL algorithms when data is limited. Hessel et al. (2018); Kielak (2020); van Hasselt et al. (2019) show combining existing RL algorithms (Rainbow) can boost data efficiency. Srinivas et al. (2020) proposed to combine contrastive loss with image augmentation while follow-up results from Laskin et al. (2020) suggest that the most of the benefits come from its use of image augmentation. Laskin et al. (2020); Kostrikov et al. (2020) demonstrate applying modest image augmentation can substantially improve data efficiency in vision-based RL. Our work improves data efficiency of RL in an orthogonal direction by unsupervised pre-training, the above advances can be used inside of APT to obtain better RL optimization in both pre-training and fine-tuning phases.

## 3 METHOD

Our method shown in Figure 2 consists of contrastive representation learning and particle-based entropy maximization in the learned representation space. Consider an agent that sees an observation $x_t$, takes an action $a_t$ and transitions to the next state with observation $x_{t+1}$ following unknown environmental dynamics. We want to incentivize this agent with a reward $r_t$ relating to how informative the transition was. The goal of the agent is to learn a good representation $f_\theta(x)$ of the observation $x$, or a good initialization of policy $\pi(a|x)$ by interacting with the environment in a reward-free manner, such that fine-tuning on downstream tasks achieves higher long-term cumulative task-specific reward than training from scratch.

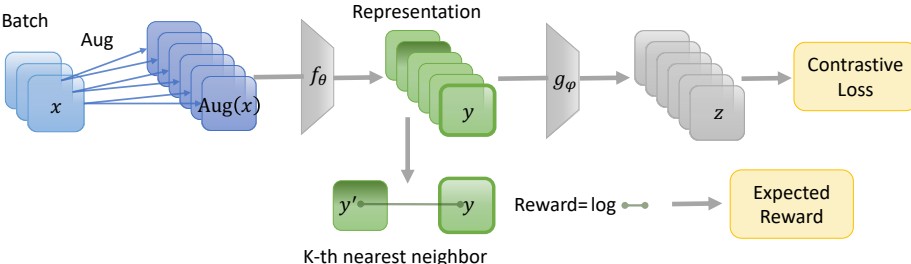

Figure 2: Diagram of the proposed method Unsupervised Active Pre-Training: it consists of contrastive representation learning on data collected by the agent (equation (1)) and RL optimization to maximize particle based entropy (equation (5)). After pre-training, the task-agnostic encoder $f_\theta$ and the RL policy initialization can be fine-tuned for different downstream tasks to maximize task-specific reward.

**Learning contrastive representations**  Within each batch of transitions sampled from the replay buffer. We apply data augmentation to each data point and the augmented observations are encoded into a small latent space where a contrastive loss is applied. Our contrastive learning is based on SimCLR (Chen et al., 2020b), chosen for its simplicity.

$$\min_{\theta,\phi} -\mathbb{E}\left[\log \frac{\exp(z_i^T z_j)}{\sum_{i=1}^{2N} \mathbb{I}_{[k\neq i]} \exp(z_i^T z_k)}\right],\qquad(1)$$

where $x$ is a $n$-dimensional data point in the observation space $\mathcal{X} \subseteq \mathbb{R}^n$, and $z_i$, $z_j$ are normalized $d_{\mathcal{Z}}$-dimensional vectors of two random augmentations $x_i$ and $x_j$ of the data point $x$ followed by a deterministic mapping $g_\phi(f_\theta(x_i)(\cdot))$, and $f_\theta$ is representation encoder given by $f_\theta : \mathbb{R}^n \to \mathbb{R}^{d_{\mathcal{Y}}}$, and $g_\phi$ is a projection head $g_\phi : \mathbb{R}^{d_{\mathcal{Y}}} \to \mathbb{R}^{d_{\mathcal{Z}}}$, and $N$ is the batch size. This objective tries to maximally distinguish an input $x_i$ from alternative inputs $x_j$. The intuition is that by doing so, the representation captures important information between similar data points, and therefore improve performance on downstream tasks.

**Particle based entropy maximization**  Let the distribution of observations be $p(x)$. The entropy of the observations is given by $H(p) = -\mathbb{E}_{x\sim p(x)}[\log p(x)]$. However, in high-dimensional spaces it is challenging to estimate the density, preventing us from directly maximizing the exact entropy.

To remedy this issue, we resort to the particle-based entropy estimator (Singh et al., 2003; Beirlant, 1997) which is based on $k$-Nearest Neighbors ($k$NN). The particle based entropy estimate is given by

$$\widehat{H}_k(p) = -\frac{1}{N}\sum_{i=1}^{N} \log \frac{k}{N\mathrm{Vol}_i^k} + \log k - \Psi(k) \propto \sum_{i=1}^{N} \mathrm{Vol}_i^k,\qquad(2)$$

where $\Psi$ is the digamma function, $\log k - \Psi(k)$ is a bias correction term, $\mathrm{Vol}_i^k$ is the volume of the hyper-sphere of radius $R_i = \|x_i - x_i^{k\mathrm{NN}}\|$, which is the Euclidean distance between $x_i$ and its k-th nearest neighbor $x_i^{k\mathrm{NN}}$. The volume is given by:

$$\mathrm{Vol}_i^k = \frac{\|x_i - x_i^{k\mathrm{NN}}\|_n^n \cdot \pi^{n/2}}{\Gamma\left(\frac{p}{2} + 1\right)}, \tag{3}$$

where $\Gamma$ is the gamma function, and $n$ the dimensions of $\mathcal{X}$. Put equation (2) and equation (3) together, we simplify the entropy:

$$\widehat{H}_k(p) \propto \sum_{i=1}^N \log \|x_i - x_i^{k\mathrm{NN}}\|_n^n. \tag{4}$$

When the target data distribution $p'(x)$ (as in the case of off-policy RL, which we use for sample efficiency) is different from the sampling distribution $p(x)$, in principle importance sampling is needed to correct bias (Ajgl & Šimandl, 2011). However, empirically we find the biased approximation in equation (2) works fine and does not need to estimate the high variance importance ratios, as shown in Section B (Appendix).

Given a batch of transitions sampled from the replay buffer, we associate the particle based entropy estimation with the intrinsic reward, and use off-policy RL algorithm to maximize the expected reward. With the objective of particle based entropy given in equation (4), we might able to maximize state entropy in continuous control, but it is still not applicable to learn visual RL agents in high-dimensional domains like DMControl and Atari games.

To remedy this issue, we maximize the entropy in our learned lower-dimensional representation space, we do this by jointly learning representations by contrastive learning equation (1) and exploring by particle based entropy maximization equation (4). Specifically, for a batch of transitions $\{(x_t, a_t, x_{t+1})\}$ sampled from the replay buffer, each $x_{t+1}$ is treated as a particle and we associate each transition with a intrinsic reward given by

$$r(x_t, a_t, x_{t+1}) := \log(\|y_{t+1} - y_{t+1}^{k\mathrm{NN}}\|_n^n + c), \tag{5}$$

where $y = f_\theta(x)$ is the representation (i.e. we estimate the entropy in the latent space), $c$ is a constant for numerical stability (fixed to 1 in all our experiments). In order to keep the rewards on a consistent scale, we normalized the intrinsic reward by dividing it by a running estimate of the standard deviation of the intrinsic reward. A detailed computation of the intrinsic reward in PyTorch can be found in Algorithm 2.

With the intrinsic reward defined in equation (5), we can derive the intrinsic reward decreases to 0 as most of the state space is visited which is a favorable property for pre-training.

**Lemma 1.** *Assume we have an episodic MDP setting, and a finite state space $\mathcal{X} \subseteq \mathbb{R}^n$, and a buffer of observed states $(x_1, \ldots, x_T)$ with total sample size $T$, a deterministic representation encoder $f_\theta : \mathbb{R}^n \to \mathbb{R}^{d_\mathcal{Y}}$, an intrinsic reward defined as equation (5) with $k \in \mathbb{N}$, and an optimal policy that maximize the intrinsic rewards. We can derive the intrinsic reward is 0 in the limit of sample size*

$$\lim_{T\to\infty} r(x, a, x') = 0, \ \forall x \in \mathcal{X}. \tag{6}$$

---

**Algorithm 1** Unsupervised Active Pre-Training

f: encoder network shared between actor & critic
g: projection network for contrastive learning
k: hyperparameter of $k$NN
**while** iterations so far < max pre-training iterations **do**

    Interact with a *reward-free* environment
    *// sample a batch of transitions from replay buffer*
    *// update f and g to minimize contrastive loss*
    Update(f.param, g.param)
    *// compute intrinsic reward for each transition*
    Compute $y_{t+1}$ for each sampled transition
    Set $r(x_t, a_t, x_{t+1}) = \log(\|y_{t+1} - y_{t+1}^{k\mathrm{NN}}\|_2^2 + 1)$
    *// update RL agent to maximize expected reward*
    Update(actor.param, critic.param)
**end while**

---

*Proof.* Since the intrinsic reward $r(x, a, x')$ defined in equation (5) depends on the $k$-th nearest neighbor in latent space and the encoder $f_\theta$ is deterministic, we just need to prove the visitation count $c(x)$ of $x$ is larger than $k$ as $T$ goes infinity. We know the MDP is episodic, therefore as $T \to \infty$, all states communicate and $c(x) \to \infty$, thus we have $\lim_{T\to\infty} c(x) \geq k, \forall k \in \mathbb{N}, \forall x \in \mathcal{X}$. $\qquad \square$

While the assumption of finite state space may not be true for large complex environment like Atari games, lemma 1 gives more insights on using this particular intrinsic reward for pre-training. We use

$n = 2$ in our implementation because of the structure we imposed on the contrastive representations. The $k$NN in principle should be computed over the entire buffer which means it scales linearly with sample size, we instead compute the intrinsic reward within current batch to trade-off computation. We fixed $k = 3$ in all our experiments as we found it works well in initial experiments. APT alternates between minimizing contrastive loss in equation (1) and maximizing expected intrinsic reward in equation (5). The pseudocode of APT is shown in Algorithm 1 and the full pseudocode of the algorithm is summarized in Algorithm 3 (Appendix). The diagram of APT is shown in Figure 2.

## 4 EXPERIMENTS

### 4.1 EXPERIMENTAL SETUP

The evaluation benchmarks are DeepMind Control Suite (DMControl; Tassa et al., 2020) and Atari suite (Bellemare et al., 2013) from OpenAI Gym (Brockman et al., 2016). For DMControl, we use pixel observation instead of state as input. For all of the experiments, we use DrQ as the underlying RL optimization algorithm. Unless stated otherwise, all curves are the average of three runs with different seeds, and the shaded areas are standard errors of the mean. The results reported on Atari games suite are averaged over five runs with different seeds. In addition to the existing tasks in DMControl, we also design several new sparse reward tasks: (1) *{HalfCheetah, Hopper, Walker} Jump Sparse*: the agent receives a positive reward 1 for jumping above a given height otherwise reward is 0. (2) *{HalfCheetah, Hopper, Walker} Reach Sparse*: the agent receives positive reward 1 for reaching a given target location otherwise reward is 0. (3) *Walker Escape Sparse*: the initial position of Walker is turned upside down, and receives reward 1 for successfully turning itself over otherwise 0. In all the considered tasks, the episode ends when the goal is reached.

Table 1: Evaluation on Atari games. $@N$ represents the amount of RL interaction utilized. $Mdn$ is the median of human-normalized scores, $M$ is the mean, $> 0$ is the number of games with better than random performance, and $> H$ is the number of games with human-level performance. On each subset, we mark as bold the highest score. Since different papers report different results of supervised RL $e.g$ SimPLe, we choose the best available results and contrast them to APT's results. The results of VISR are cited from Hansen et al. (2020) as the source code is not publicly available. Raw scores of each Atari game given Table 5 (Appendix). **Top**: data-limited RL. **Bottom**: RL with unsupervised pre-training.

| Algorithm | 26 Game Subset | | | | 47 Game Subset | | | | Full 57 Games | | | |
|---|---|---|---|---|---|---|---|---|---|---|---|---|
| | Mdn | M | $>0$ | $>H$ | Mdn | M | $>0$ | $>H$ | Mdn | M | $>0$ | $>H$ |
| *// Fully-supervised training* | | | | | | | | | | | | |
| SimPLe @100$k$ | 14.39 | 44.30 | **26** | 2 | – | – | – | – | – | – | – | – |
| OTRainbow @100$k$ | 20.40 | 26.42 | **26** | 1 | – | – | – | – | – | – | – | – |
| DrQ @100$k$ | 28.42 | 35.70 | 24 | 2 | – | – | – | – | – | – | – | – |
| PPO @500$k$ | 20.93 | 43.74 | 25 | **7** | – | – | – | – | – | – | – | – |
| DQN @10$M$ | 27.80 | 52.95 | 25 | **7** | 9.91 | 28.07 | **41** | 7 | 8.61 | 27.55 | **48** | 7 |
| *// Unsupervised pre-training with supervised fine-tuning @100$k$* | | | | | | | | | | | | |
| DIAYN | 0.01 | 16.94 | 13 | 2 | 1.31 | 19.64 | 28 | 6 | 1.55 | 16.65 | 33 | 6 |
| VISR | 9.50 | **128.07** | 21 | 7 | 9.42 | 121.08 | 35 | 11 | 6.81 | 102.31 | 40 | 11 |
| GPI VISR | 6.59 | 111.23 | 22 | 7 | 11.70 | **129.76** | 38 | 12 | 8.99 | **109.16** | 44 | 12 |
| Count-based | 1.23 | 21.94 | 16 | 3 | – | – | – | – | – | – | – | – |
| APT (ours) | **38.23** | 59.89 | **26** | 7 | **34.22** | 61.78 | 36 | **12** | **41.25** | 87.33 | 43 | **12** |

### 4.2 SINGLE ENVIRONMENT PRE-TRAINING

We conduct a full evaluation of APT on a diverse set of tasks in the single environment setting, the models are pre-trained on Cheetah, Hopper, and Walker for a long period of interacting without reward supervision ($5M$ steps), then fine-tuned for 15 downstream RL tasks such as controlling the Walker to turn itself over. Fine-tuning representation is a common practice in deep learning (Krizhevsky et al., 2012; He et al., 2016). In our RL experiments, APT stands for fine-tuning both representation and RL agent initialization. The main baseline is count-based exploration (Bellemare et al., 2014; 2016; Ostrovski et al., 2017), which was proposed as a way to estimate counts in high dimensional

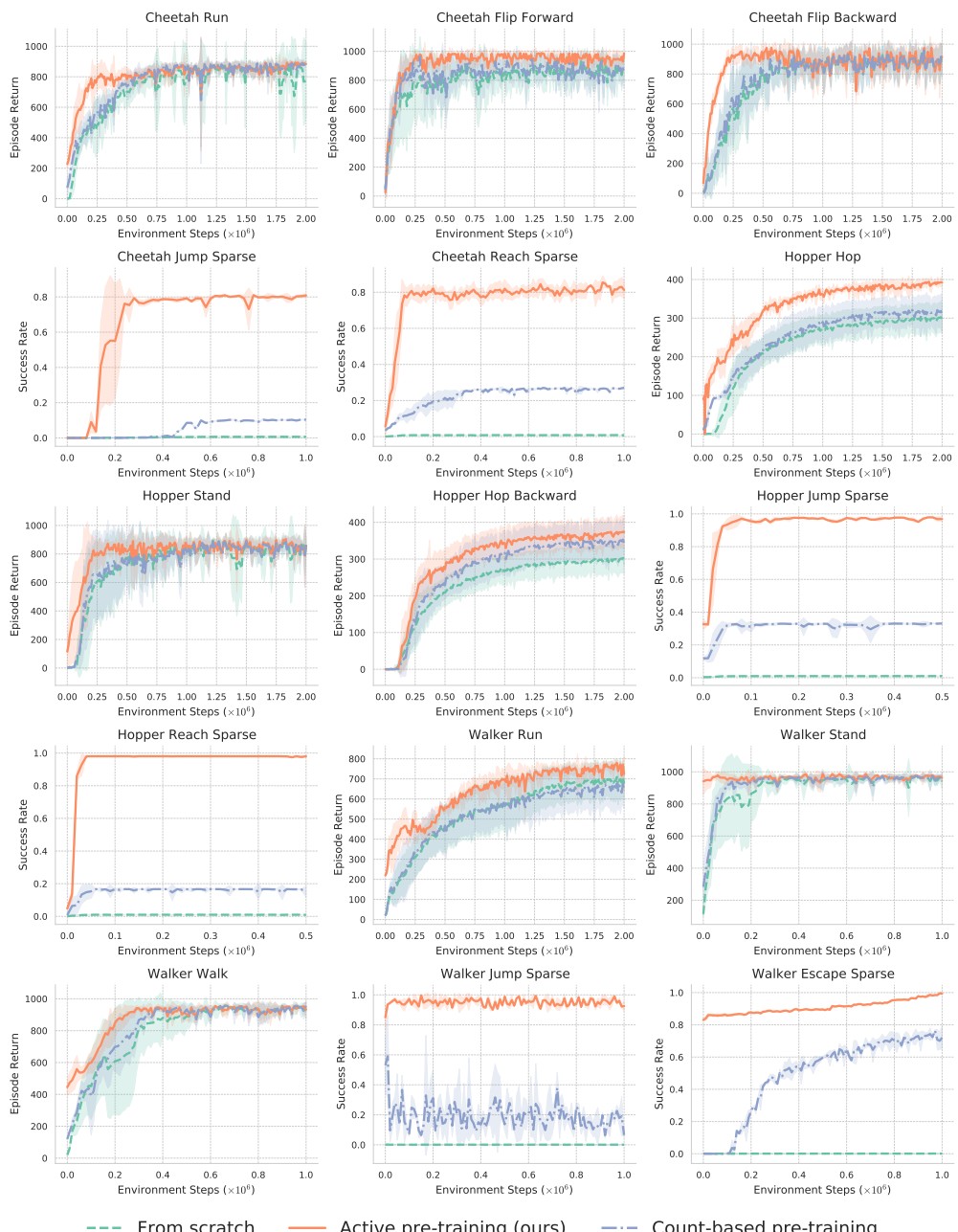

Figure 3: Evaluation on DeepMind Control suite. Models are pre-trained on Cheetah, Hopper, and Walker, and subsequently fine-tuned on respective downstream tasks. The 'sparse' denotes reward is sparse. The scores of each environment given in Table 4 (Appendix).

states spaces by estimating density; the agent is then encouraged to visit states with a low visit count. We followed Ostrovski et al. (2017) to use PixelCNN (Van den Oord et al., 2016) as a density model. Count-based exploration with PixelCNN is an efficient exploration method in sparse reward environments (Taïga et al., 2019), we will refer to this baseline as count-based exploration or count-based throughout the rest of the paper.

Results on DMControl shown in Figure 3 demonstrate that APT beats all baselines on all the tasks across different environments, while the sparse reward tasks are extremely difficult for training from scratch. In some cases, APT allows for very rapid fine-tuning, indicating APT learns reward-free representation and meaningful RL initialization. The significantly superior performance of APT empirically shows that maximizing particle-based entropy can drive the RL agent to collect diverse samples and learn reward-free initializations that are effective for downstream tasks.

The evaluation on the full suite of 57 Atari games (Bellemare et al., 2013) follows the setting of VISR (Hansen et al., 2020). Firstly, we evaluate APT in a two-phase setup. Agents are allowed a long unsupervised pre-training phase without access to rewards, followed by a short test phase(100k steps). DIAYN (Eysenbach et al., 2018) is a skill-discovery method that maximizes the mutual information between latent variable polices and their behavior in terms of state visitation. The main baseline is VISR (and its variant GPI VISR), which combines skill discovery with universal successor approximators (Borsa et al., 2018) to enable fast task inference at both training and test phases (Barreto et al., 2017; 2018). Due to the high computational cost, the count-based baseline is not evaluated on the entire 57 games suite but the 26 games subset (Kaiser et al., 2019). Secondly, we contrast APT with canonical RL algorithms in the low-data regime, following the setting of Kaiser et al. (2019). The compared algorithms include DrQ, SimPLe (Kaiser et al., 2019), proximal policy optimization (PPO) (Schulman et al., 2017), and OTRainbow (Kielak, 2020).

Results shown in Table 1 demonstrate that APT significantly outperforms all baselines and buying performance equivalent to hundreds of millions of sampling steps. Note that it is possible we can further improve the performance by directly applying VISR on top of the pre-trained models learned by APT, we leave it for future direction. A further discussion of the connections and differences between APT and VISR can be found in Section C in Appendix which gives more intuitions.

### 4.3 REPRESENTATION AND RL AGENT INITIALIZATION FINE-TUNE

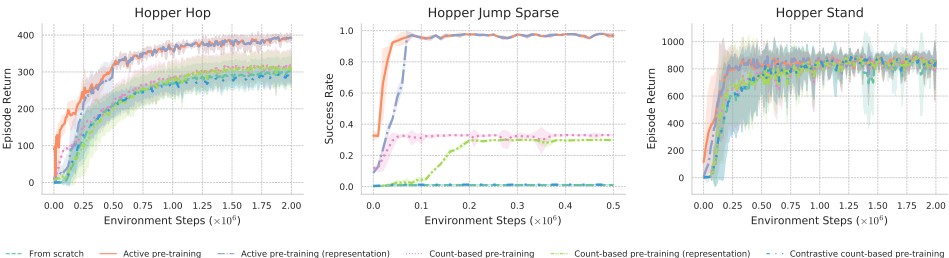

Figure 4: Comparison of fine-tuning representation, fine-tuning both representation and RL agent, and ablated baselines. Models are pre-trained on Hopper and subsequently fine-tuned on downstream tasks. The 'sparse' denote reward is sparse. Both variants of APT outperform training from scratch and other baselines.

Table 2: Comparison of fine-tuning representation, fine-tuning both representation and RL agent on Atari games. The results are obtained after fine-tuning $100k$ timesteps and are averaged over five random seeds.

| Atari Games | Human | Random | VISR | APT(representation) | APT (ours) |
|---|---|---|---|---|---|
| *// Hard exploration games (dense reward)* | | | | | |
| BanHeist | 753.1 | 14.2 | $200.3 \pm 14.9$ | $347.7 \pm 24.6$ | $\mathbf{456.7 \pm 15.6}$ |
| Hero | 30826.4 | 1027.0 | $663.5 \pm 31.8$ | $4571.6 \pm 234.6$ | $\mathbf{6789.1 \pm 171.4}$ |
| *// Hard exploration games (sparse reward)* | | | | | |
| Freeway | 29.6 | 0.0 | $-2.1 \pm 0.5$ | $12.9 \pm 0.7$ | $\mathbf{29.9 \pm 1.3}$ |
| MontezumaRevenge | 4753.3 | 0.0 | $0.0 \pm 0.0$ | $43.1 \pm 15.6$ | $\mathbf{147.0 \pm 24.3}$ |

We investigated the difference between fine-tuning only the representation and fine-tuning both representation and policy initialization. APT denotes fine-tuning both representation and RL policy initialization while APT (representation) stands for fine-tuning encoder from pre-trained models and randomly initializing policy. The notable difference is that APT (representation) decouples the action space dimension from pre-trained models. We applied count-based exploration on top of contrastive learning representations to eliminate the potential effect of representation learning difference, this baseline is shown as contrastive count-based pre-training, where we train VAE (Kingma & Welling, 2013) on the learned contrastive representations to estimate state visitation.

As we show in Figure 4, APT (representation) beats all of the supervised RL and pre-training RL baselines, fine-tuning both representation and RL initialization further improve performance. In some cases, APT allows for more rapid performance improvement than APT (representation) in a small fraction of the total number of samples, indicating APT learns meaningful reward-free and task-agnostic behavior. Both APT and APT (representation) significantly outperform the ablated

contrastive count-based pre-training baseline, confirming the particle-based entropy maximization is a crucial component.

Table 2 shows the results of fine-tuning pre-trained models for $100k$ timesteps on Atari games. VISR tends to be more data efficient than APT and APT(representation) on easy exploration games, potentially due to the explicit reward regression and successor feature in VISR. On hard exploration games, e.g., Freeway, APT has a significant advantage, achieving an order of magnitude higher scores than VISR while maintaining a very high score across the remaining games. APT significantly outperforms APT(representation) on Atari games, showing the learned exploratory policy is crucial for learning with a very limited number of interactions.

## 4.4 DISCUSSION OF MULTI-ENVIRONMENT PRE-TRAINING

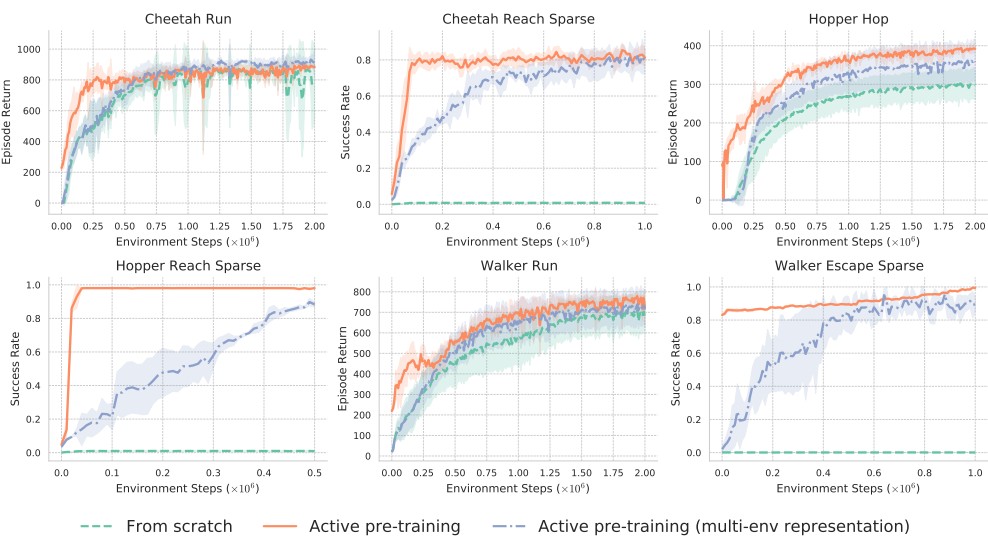

Figure 5: Comparison between fine-tuning representations learned on single environment and multiple environments. We apply APT to all the three environments.

We completed an initial exploration of APT (representation) in multi-environment setting on DMControl suite, results shown in Table 5. In this setup, APT simultaneously learns pre-trained representation from Hopper, Cheetah, and Walker environments. We evaluate the pre-trained model by using it in separate RL agents learning each downstream task from every environment. As shown in Figure 5, we found multi-environment pre-training outperforms training from scratch in sparse reward tasks and performs on par or better than training from scratch in dense reward tasks, showing multi-environment pre-training is efficient. Comparing with single environment APT, multi-environment pre-training tends to have mixed results, indicating there exists intervention between representation learning in different environments. We remark despite the multi-environment variant of APT is outperformed by APT, the multi-environment pre-training is a novel research direction and our careful implementation considerations and extensive experimental results allow the method to be widely adopted.

## 5 CONCLUSION

A new unsupervised pre-training method for RL is introduced to address reward-free pre-training for visual RL. On DMControl suite and Atari games, our method dramatically improves performance on tasks that are extremely difficult for training from scratch. Our method achieves the results of fully supervised canonical RL algorithms using a small fraction of total samples and outperforms data-efficient supervised RL methods. Our major contribution is proposing an efficient algorithm for maximizing particle-based entropy in the latent representation space, allowing the same task-agnostic pre-trained model to successfully tackle a broad set of RL tasks.

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

# A   REWARD COMPUTATION IN PYTORCH

---

**Algorithm 2** Computation of reward in APT

---

```python
import torch

device = torch.device("cuda" if torch.cuda.is_available() else "cpu")

class RMS(object):
    def __init__(self, epsilon=1e-4, shape=(1,)):
        self.M = torch.zeros(shape).to(device)
        self.S = torch.ones(shape).to(device)
        self.n = epsilon

    def __call__(self, x):
        bs = x.size(0)
        delta = torch.mean(x, dim=0) - self.M
        new_M = self.M + delta * bs / (self.n + bs)
        new_S = (self.S * self.n + torch.var(x, dim=0) * bs +
        torch.square(delta) * self.n * bs / (self.n + bs)) / (self.n + bs)

        self.M = new_M
        self.S = new_S
        self.n += bs

        return self.M, self.S

rms = RMS()   # moving statistics of reward
# source: batch of states sampled from replay buffer (b1, c)
# target: same as source or all of the states in replay buffer (b2, c)
# encoder: contrastive representation encoder
def compute_intrinsic_reward(source, target, encoder, k):
    with torch.no_grad():
        source, target = encoder(source), encoder(target)
        sim_matrix = torch.norm(source[:, None, :] - target[None, :, :],
        dim=-1, p=2)   # (b1, 1, c) - (1, b2, c) -> (b1, b2, c) -> (b1, b2)
        sim_weight, sim_indice = sim_matrix.topk(k, dim=1, largest=False)  #
        (b1, k)
        reward = torch.norm(source - target[sim_indice[:, -1]], dim=1, p=2,
        keepdim=True)   # (b1, k)   # k-th based
        moving_mean, moving_std = rms(reward)
        reward = reward / moving_std
        reward = torch.log(reward + 1.0)
    return reward
```

---

## B    EVALUATION OF ENTROPY MAXIMIZATION

We conducted experiments to evaluate APT's performance in maximizing entropy in state based continuous control tasks from OpenAI Gym (Brockman et al., 2016). We compared APT with state-of-the-art state entropy maximization methods, including (1) MaxEnt (Hazan et al., 2019) which proposes a provable efficient exploration method for maximizing state entropy, and shows their method can improve the efficiency of exploring the state space in continuous control. (2) MEPOL (Mutti et al., 2020) which

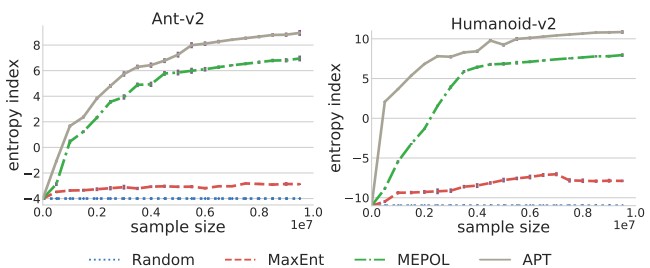

Figure 6: Comparison of entropy between different state exploration methods. The results are averaged over 5 random trials. Error bar denotes one standard derivation.

is a recent state-of-the-art of exploration in continuous control, their estimation of particle based entropy based on importance sampling and is unbiased. (3) APT-like MEPOL which denotes training APT objective with MEPOL's trust region optimization.

Following MaxEnt (Hazan et al., 2019), we compute the entropy index by discretizing the state space. The Ant-v2 Jump and Humanoid-v2 Standup tasks are two sparse reward tasks given in MEPOL (Mutti et al., 2020).

Figure 6 shows the results of entropy achieved by different methods, Table 3 and shows the results of fine-tuning pre-trained models for $0.5M$ timesteps. MaxEnt performs poorly in high-dimensional continuous control due to state density modeling is difficult. APT-like MEPOL outperforms MEPOL, indicating the objective of APT balances between variance and bias. APT significantly outperforms all baselines in maximizing entropy and also has the highest data efficiency in solving sparse reward tasks, confirming that the off-policy nature of APT is crucial for both pre-training and fine-tuning.

| Gym Environments | From scratch (state input) | MEPOL | APT-like MEPOL | MaxEnt | APT (ours) |
|---|---|---|---|---|---|
| *Sparse reward tasks* | | | *Success rate* | | |
| Ant-v2 Jump | $.12 \pm .07$ | $.44 \pm .29$ | $.52 \pm .04$ | $.18 \pm .05$ | $\mathbf{.96 \pm .10}$ |
| Humanoid-v2 Standup | $.0 \pm .0$ | $.72 \pm .21$ | $.78 \pm .06$ | $.12 \pm .04$ | $\mathbf{.94 \pm .13}$ |
| *Dense reward tasks* | | | *Episode reward* | | |
| Ant-v2 | $4278.1 \pm 712.7$ | $1467.3 \pm 879.4$ | $1635.7 \pm 876.9$ | $1534.6 \pm 318.6$ | $\mathbf{4857.8 \pm 875.9}$ |
| Humanoid-v2 | $4854.3 \pm 1012.6$ | $1427.3 \pm 567.8$ | $1987.4 \pm 467.8$ | $678.4 \pm 267.8$ | $\mathbf{5036.9 \pm 1367.8}$ |

Table 3: Comparison of fine-tuning on continuous control environments in OpenAI Gym. Maximum value for each task is bolded. $\pm$ corresponds to a single standard deviation over 3 runs with random seed.

## C    ANALYSIS VARIATIONAL UNSUPERVISED RL METHODS

We contrast APT with variational unsupervised RL algorithms to give more intuitions besides the empirical evaluation in Table 1.

Utilizing a strong inductive bias that is likely to yield features relevant to rewards of possible downstream tasks has been the central goal of unsupervised RL research. One of the widely used such bias is proposed by Achiam & Sastry (2017); Gregor et al. (2016) that is to only represent the subset of observation space that the agent can control. This can be accomplished by maximizing the mutual information between a policy conditioning variable and the agent's behavior. Formally, the goal is to learn latent-conditioned policies $\pi_\theta(a|s, z)$ and define *skills* as the policies obtained when conditioning $\pi$ on a fixed value of $z \in Z$. There exist many algorithms that optimize policy parameter $\theta$ maximize the mutual information through various means (see *e.g.* Eysenbach et al., 2018; Hansen et al., 2020; Warde-Farley et al., 2018; Sharma et al., 2019). The quantity can be derived by

expanding the definition and derive a variational lower bound (Barber & Agakov, 2003).

$$J(\theta) = I(z; s) \tag{7}$$
$$= H(z) - H(z|s) \tag{8}$$
$$= \mathbb{E}_{\pi,z}[\log q(z|s)] + \mathbb{E}_s[KL(p(\cdot|s)||q_\phi(\cdot|s))] + H(z) \tag{9}$$
$$\geq \mathbb{E}_{\pi,z}[\log q_\phi(z|s)] + H(z), \tag{10}$$
$$\tag{11}$$

where $q_\phi(z|s)$ is a variational approximation. In practice, sampling $z$ from a fixed distribution yields better and more stable results (Eysenbach et al., 2018; Hansen et al., 2020), which simplifies the objective to maximizing the conditional entropy,

$$L(\theta, \phi) = \mathbb{E}_{\pi,z}[\log q_\phi(z|s)]. \tag{12}$$

The optimization of equation (12) is then accomplished by RL algorithm by defining reward

$$r(s, a, s') = \log q_\phi(z|s) \tag{13}$$

equation (12) has been shown effective in RL, from learning skills in state based control in DI-AYN (Eysenbach et al., 2018) and EDL (Campos et al., 2020) to combining successor features with skill discovery in VISR (Hansen et al., 2020).

Comparing the variational based intrinsic reward equation (13) with the intrinsic reward used in APT( equation (5)), we can see that the particle-based reward does not need to learn a parametric probabilistic density and thus gracefully scales to high dimensional vision-based RL. We see that APT performs considerably better on downstream RL tasks as shown in Table 1.

## D    COMPARISON ON DMCONTROL

| DMC Environment | From scratch (visual) | From scratch (state) | Count-based | DIAYN | VISR | APT (ours) |
|---|---|---|---|---|---|---|
| *//Dense reward tasks* | *Mean episode return* | | | | | |
| Cheetah Run | $660 \pm 96$ | $\mathbf{772 \pm 60}$ | $610 \pm 78$ | $256 \pm 128$ | $640 \pm 134$ | $671 \pm 89$ |
| Cheetah Flip Forward | $885 \pm 112$ | $937 \pm 73$ | $867 \pm 98$ | $432 \pm 78$ | $897 \pm 215$ | $\mathbf{987 \pm 97}$ |
| Cheetah Flip Backward | $879 \pm 87$ | $\mathbf{923 \pm 145}$ | $871 \pm 79$ | $465 \pm 89$ | $868 \pm 187$ | $856 \pm 103$ |
| Hopper Hop | $314 \pm 167$ | $287 \pm 145$ | $346 \pm 198$ | $389 \pm 211$ | $353 \pm 189$ | $\mathbf{397 \pm 101}$ |
| Hopper Stand | $845 \pm 103$ | $\mathbf{859 \pm 97}$ | $834 \pm 121$ | $821 \pm 109$ | $798 \pm 123$ | $832 \pm 112$ |
| Hopper Hop Backward | $290 \pm 97$ | $381 \pm 109$ | $361 \pm 81$ | $267 \pm 65$ | $376 \pm 54$ | $\mathbf{387 \pm 112}$ |
| Walker Run | $713 \pm 139$ | $\mathbf{787 \pm 37}$ | $683 \pm 151$ | $697 \pm 63$ | $675 \pm 71$ | $765 \pm 63$ |
| Walker Stand | $987 \pm 40$ | $991 \pm 29$ | $961 \pm 39$ | $974 \pm 78$ | $\mathbf{996 \pm 67}$ | $979 \pm 31$ |
| Walker Walk | $965 \pm 35$ | $979 \pm 67$ | $958 \pm 29$ | $945 \pm 71$ | $\mathbf{987 \pm 37}$ | $969 \pm 51$ |
| *//Sparse reward tasks* | *Mean success rate* | | | | | |
| Cheetah Jump Sparse | $.0 \pm .0$ | $.38 \pm .13$ | $.09 \pm .03$ | $.31 \pm .12$ | $.68 \pm .12$ | $\mathbf{.79 \pm .13}$ |
| Cheetah Reach Sparse | $.0 \pm .0$ | $.18 \pm .06$ | $.23 \pm .08$ | $.53 \pm .15$ | $\mathbf{.93 \pm .16}$ | $.83 \pm .12$ |
| Hopper Jump Sparse | $.0 \pm .0$ | $.53 \pm .14$ | $.37 \pm .08$ | $.51 \pm .23$ | $.87 \pm .12$ | $\mathbf{.91 \pm .05}$ |
| Hopper Reach Sparse | $.0 \pm .0$ | $.37 \pm .08$ | $.16 \pm .07$ | $.65 \pm .18$ | $.93 \pm .06$ | $\mathbf{.94 \pm .04}$ |
| Walker Jump Sparse | $.0 \pm .0$ | $.13 \pm .02$ | $.05 \pm .01$ | $.14 \pm .08$ | $.68 \pm .12$ | $\mathbf{.87 \pm .09}$ |
| Walker Escape Sparse | $.0 \pm .0$ | $.09 \pm .02$ | $.67 \pm .14$ | $.95 \pm .17$ | $.87 \pm .12$ | $\mathbf{.97 \pm .03}$ |

Table 4: Comparison of fine-tuning on DMControl. Models are pre-trained on Cheetah, Hopper, and Walker, and subsequently fine-tuned on respective downstream tasks. The 'sparse' denotes reward is sparse. APT significantly outperforms baselines in most of sparse reward tasks.

# E   COMPARISON ON ATARI GAMES

Table 5: Comparison of raw scores of each method on Atari games. Results are averaged over five random seeds. ■ denotes dense reward hard exploration games. ■ denotes sparse reward hard exploration games. @$N$ represents the amount of RL interaction utilized at fine-tuning phase.

| Atari game | Human | Random | VISR@100$k$ | APT@0 | APT@100$k$ (ours) |
|---|---|---|---|---|---|
| Alien | 7127.7 | 227.8 | 364.4 | 287.7 | **2614.8** |
| Amidar | 1719.5 | 5.8 | 286.0 | 256.8 | **1231.4** |
| Assault | 742.0 | 222.4 | **1209.1** | 345.1 | 891.5 |
| Asterix | 8503.3 | 210.0 | **6216.7** | 234.8 | 185.5 |
| Asteroids | 47388.7 | 719.1 | **4443.3** | 45.6 | 678.7 |
| Atlantis | 29028.1 | 12850.0 | **140542.8** | 3451.9 | 40231.0 |
| BankHeist | 753.1 | 14.2 | 200.3 | 267.7 | **456.7** |
| BattleZone | 37187.5 | 2360.0 | 7072.7 | 3491.8 | **7075.1** |
| BeamRider | 16826.5 | 363.9 | 1741.9 | 1348.3 | **3487.2** |
| Berzerk | 2630.4 | 123.7 | 491.4 | 120.8 | **493.4** |
| Bowling | 160.7 | 23.1 | **21.2** | -16.9 | -56.5 |
| Boxing | 12.1 | 0.1 | **43.0** | 33.7 | 21.3 |
| Breakout | 30.5 | 1.7 | 397.9 | 231.8 | **480.9** |
| Centipede | 12017.1 | 2090.9 | **7184.9** | 3678.1 | 6233.9 |
| ChopperCommand | 7387.8 | 881.0 | **800.8** | 98.7 | 317.0 |
| CrazyClimber | 35829.4 | 10780.5 | **49373.9** | 1897.9 | 4128.0 |
| Defender | 18688.9 | 2874.5 | **15876.1** | 1154.3 | 5927.9 |
| DemonAttack | 1971.0 | 152.1 | 8994.9 | 945.8 | 7771.8 |
| DoubleDunk | -16.4 | -18.6 | -22.6 | **-10.8** | -17.2 |
| Enduro | 860.5 | 0.0 | -3.1 | -10.9 | **-0.3** |
| FishingDerby | -38.7 | -91.7 | -93.9 | -5.9 | **-1.6** |
| Freeway | 29.6 | 0.0 | -2.1 | 12.1 | **29.9** |
| Frostbite | 4334.7 | 65.2 | 230.9 | 771.3 | **2196.1** |
| Gopher | 2412.5 | 257.6 | 1298.6 | 1298.6 | **8190.4** |
| Gravitar | 3351.4 | 173.0 | 328.1 | 228.6 | **542.0** |
| Hero | 30826.4 | 1027.0 | 663.5 | 363.9 | **6789.1** |
| IceHockey | 0.9 | -11.2 | -18.1 | **-16.4** | -30.1 |
| Jamesbond | 302.8 | 29.0 | **484.4** | 10.3 | 356.1 |
| Kangaroo | 3035.0 | 52.0 | **1761.9** | 871.9 | 412.0 |
| Krull | 2665.5 | 1598.0 | **3142.5** | 849.8 | 2312.0 |
| KungFuMaster | 22736.3 | 258.5 | 16754.9 | 3871.9 | **17357.0** |
| MontezumaRevenge | 4753.3 | 0.0 | 0.0 | 1.3 | **147.0** |
| MsPacman | 6951.6 | 307.3 | 558.5 | 403.5 | **2427.1** |
| NameThisGame | 8049.0 | 2292.3 | **2605.8** | 698.5 | 1387.2 |
| Phoenix | 7242.6 | 761.4 | **7162.23** | 2871.9 | 3874.2 |
| Pitfall | 6463.7 | -229.4 | -370.8 | -54.9 | **0.8** |
| Pong | 14.6 | -20.7 | -26.2 | -21.8 | **-8.0** |
| PrivateEye | 69571.3 | 24.9 | 428.3 | -54.9 | **556.1** |
| Qbert | 13455.0 | 163.9 | 666.3 | 459.8 | **17671.2** |
| Riverraid | 17118.0 | 1338.5 | **5422.2** | 3871.9 | 4671.0 |
| RoadRunner | 7845.0 | 11.5 | **6146.7** | 987.9 | 4782.1 |
| Robotank | 11.9 | 2.2 | 10.0 | 9.1 | **13.7** |
| Seaquest | 42054.7 | 68.4 | 706.6 | 1891.4 | **2116.7** |
| Skiing | -4336.9 | -17098.1 | **-4312.4** | -29819.4 | -38434.1 |
| Solaris | 12326.7 | 1236.3 | 841.5 | 367.9 | **1925.8** |
| SpaceInvaders | 1668.7 | 148.0 | **9741.0** | 1389.7 | 3687.2 |
| StarGunner | 10250.0 | 664.0 | **25827.5** | 109.8 | 8717.0 |
| Surround | 6.5 | -10.0 | -15.5 | -5.9 | **-4.5** |
| Tennis | -8.3 | -23.8 | 0.7 | -8.7 | **1.2** |
| TimePilot | 5229.2 | 3568.0 | **4503.6** | 968.1 | 1567.0 |
| Tutankham | 167.6 | 11.4 | 50.7 | 78.8 | **124.6** |
| UpNDown | 11693.2 | 533.4 | **17037.6** | 81.1 | 8289.4 |
| Venture | 1187.5 | 0.0 | -1.7 | 13.8 | **231.0** |
| VideoPinball | 17667.9 | 0.0 | **35120.3** | 989.5 | 2817.1 |
| WizardOfWor | 4756.5 | 563.5 | 1453.3 | 746.9 | **3465.0** |
| YarsRevenge | 54576.9 | 3092.9 | **5543.5** | 719.3 | 1871.5 |
| Zaxxon | 9173.3 | 32.5 | 1092.5 | 518.6 | **5431.0** |

# F EXPERIMENT DETAILS

## F.1 IMAGENET BASED PRE-TRAINING

In Figure 1, the ImageNet (Deng et al., 2009) pre-trained model is based on MoCo (He et al., 2019). The policies are represented by the Impala convolutional residual network as in (Espeholt et al., 2018), with the LSTM (Hochreiter & Schmidhuber, 1997) part excluded. Images sampled from ImageNet are downsampled to $84 \times 84$, followed by frame-stacking and data augmentation. We experiment with the data augmentation methods used in He et al. (2019); Chen et al. (2020c) and the simpler random crop used in RL (Kostrikov et al., 2020). Results on DMControl shown no benefit comes from ImageNet unsupervised pre-trained models. To improve the quality of pre-trained representations, we consider initializing the filters in the first layer with weights from the model of He et al. (2016) which is trained on ImageNet classification. Similar to fine-tuning MoCo trained models, we observe no difference in performance between fine-tuning supervised pre-trained models observed and training from scratch.

## F.2 GENERAL IMPLEMENTATION DETAILS

The encoder network $f$ is a `ReLU` convolution neural network followed by a full-connected layer normalized by `LayerNorm` (Ba et al., 2016) and a `tanh` nonlinearity applied to the output of fully-connected layer. The data augmentation is a simple random shift which has been shown effective in visual domain RL in DrQ (Kostrikov et al., 2020) and RAD (Laskin et al., 2020). Specifically, the images are padded each side by 4 pixels (by repeating boundary pixels) and then select a random $84 \times 84$ crop, yielding the original image. This procedure is repeated every time an image is sampled from the replay buffer. The learning rate of contrastive learning is $0.001$, the temperature is $0.1$. We incorporate the memory mechanism (with a moving average of weights for stabilization) from He et al. (2019); Chen et al. (2020c). We use DrQ as the RL optimization algorithm at both pre-training phase and fine-tuning phase. The batch size of contrastive learning is $1024$, the batch size of RL optimization is $512$. The pre-training phase consists of $5M$ environment steps on DMControl and $250M$ environment steps on Atari games. The replay buffer size is $100K$. The projection network is a two-layer MLP with hidden size of $128$ and output size of $64$. All hyperparameters are included in Table 7 and Table 8.

## F.3 MULTI-ENVIRONMENT PRE-TRAINING DETAILS

For the experiments of multi-environment pre-training (results shown in Figure 5), we use one separate replay buffer for each of the three environments, and compute environment specific loss using data sampled from its own replay buffer. The contrastive loss and RL loss are then summations of every environment specific loss.

## F.4 DMCONTROL HYPERPARAMETERS

| Environment name | Action repeat |
|---|---|
| Cheetah | 4 |
| Walker | 2 |
| Hopper | 2 |

Table 6: The action repeat hyper-parameter used for each environment.

## F.5 ATARI HYPERPARAMETERS

For the experiments in Atari 100k experiments, we largely reuse the hyper-parameters from DrQ (Kostrikov et al., 2020). The evaluation is done for $125k$ environment steps at the end of training for $100k$ environment steps.

---

**Algorithm 3** Pseudo-code of Unsupervised Active Pre-Training.

---

f: encoder network shared between actor & critic
g: projection network for contrastive learning
k: hyperparameter of kth-nearest neighbor

*// Unsupervised pre-training phase: learning pre-trained actor-critic*
**while** iterations so far $<$ max pre-training iterations **do**
    Interact with a *reward-free* environment and collect transitions and save to replay buffer
    *// sample a batch of transitions from replay buffer and compute contrastive loss*
    Compute contrastive_loss using equation (1)
    *// update f and g to minimize contrastive loss*
    Contrastive_loss.backward()
    Update(f.param, g.param)
    *// compute intrinsic reward for each transition*
    Compute representation $y_{t+1}$ for each $x_{t+1}$ for computing $k$NN.
    Compute $r(x_t, a_t, x_{t+1}) = \log(\|y_{t+1} - y_{t+1}^{k\text{NN}}\|_2^2 + 1)$ (equation (5))
    *// optimize reward via off-policy RL*
    Compute expected_reward
    *// update RL agent to maximize expected reward*
    Update(actor.param, critic.param)
**end while**

*// Supervised fine-tune phase: fine-tuning both actor and critic*
// If fine-tune both representation and RL agent, initialize encoder and actor-critic from pre-trained models.
// If fine-tune representation only, initialize encoder from pre-trained models and randomly initialize actor and critic.
**while** iteration number $\leq$ max fine-tune iterations **do**
    Interact with a *task-specific* environment and collect transitions and save to replay buffer
    Sample a batch of transitions
    *// fine-tune representation or fine-tune both representation and RL agent initialization*
    Update(f.param, actor.param, critic.param)
**end while**

---

| Parameter | Setting |
|---|---|
| Data augmentation | Random shifts |
| Frames stacked | 3 |
| Action repetitions | Table 6 |
| Replay buffer capacity | 100000 |
| Random steps (fine-tuning phase) | 1000 |
| RL minibatch size | 512 |
| Contrastive learning minibatch size | 512 |
| K-Nearest Neighbors K value | 3 |
| Discount $\gamma$ | 0.99 |
| RL/contrastive learning optimizer | Adam |
| RL learning rate | $10^{-3}$ |
| Contrastive learning rate | $10^{-3}$ |
| Learning rate schedule | `cosine` |
| Temperature | 0.1 |
| Shared encoder: channels | $32, 32, 32$ |
| Shared encoder: filter size | $3 \times 3, 3 \times 3, 3 \times 3$ |
| Shared encoder: stride | $2, 2, 2, 1$ |
| Actor update frequency | 2 |
| Actor log stddev bounds | $[-10, 2]$ |
| Actor: hidden units | 1024 |
| Actor: layers | 3 |
| Init temperature | 0.1 |
| Critic Q-function: hidden units | 1024 |
| Critic target update frequency | 2 |
| Critic Q-function soft-update rate $\tau$ | 0.01 |
| Non-linearity | `ReLU` |

Table 7: Hyper-parameters in the DeepMind control suite experiments.

| Parameter | Setting |
|---|---|
| Data augmentation | Random shifts and Intensity |
| Grey-scaling | True |
| Observation down-sampling | $84 \times 84$ |
| Frames stacked | 4 |
| Action repetitions | 4 |
| Reward clipping | $[-1, 1]$ |
| Terminal on loss of life | True |
| Max frames per episode | 108k |
| Update | Double Q |
| Dueling | True |
| Target network: update period | 1 |
| Discount factor | 0.99 |
| Minibatch size | 32 |
| RL optimizer | Adam |
| RL optimizer (pre-training): learning rate | 0.0001 |
| RL optimizer (fine-tuning): learning rate | 0.001 |
| RL optimizer: $\beta_1$ | 0.9 |
| RL optimizer: $\beta_2$ | 0.999 |
| RL optimizer: $\epsilon$ | 0.00015 |
| K-Nearest Neighbors K value | 3 |
| Contrastive learning optimizer | Adam |
| Contrastive learning rate | $10^{-3}$ |
| Learning rate schedule | `cosine` |
| Temperature | 0.1 |
| Max gradient norm | 10 |
| Training steps | 100k |
| Evaluation steps | 125k |
| Min replay size for sampling | 1600 |
| Memory size | Unbounded |
| Replay period every | 1 step |
| Multi-step return length | 10 |
| Q network: channels | $32, 64, 64$ |
| Q network: filter size | $8 \times 8, 4 \times 4, 3 \times 3$ |
| Q network: stride | $4, 2, 1$ |
| Q network: hidden units | 512 |
| Non-linearity | `ReLU` |
| Exploration | $\epsilon$-greedy |
| $\epsilon$-decay | 2500 |

Table 8: Hyper-parameters in the Atari suite experiments.

