# OpenReview forum: "Unsupervised Active Pre-Training for Reinforcement Learning"
_ICLR.cc/2021/Conference — Reject_

### Official Review · AnonReviewer3 · 2020-10-26
**Learning reward-free exploration via k-NN entropy estimation**

**Rating:** 5
**Confidence:** 5

**Review:**

*SUMMARY*

The paper proposes a method to simultaneously learn effective representations and efficient exploration in a reward-free context. The algorithm iterates between minimizing a contrastive loss and maximizing an intrinsic reward derived from a k-NN entropy estimation of the state distribution. Then, authors empirically evaluate the method over a set of visual Mujoco tasks and Atari games.

*STRENGTHS*
- the paper addresses a very relevant reward-free exploration objective as a preprocessing to RL
- the paper combines representation learning and state entropy maximization into a promising practical method

*WEAKNESSES*
- the paper might be too incremental with respect to previous (albeit unpublished) work
- the paper somewhat fails to empirically illustrate and validate the reward-free phase

*EVALUATION*

Unfortunately, over some concerns regarding the novelty of the presented method and its experimental validation, which I find somewhat weak for an essentially empirical work, I would lean towards rejecting the paper.

*DETAILED COMMENTS*

C1) The exploration component of APT has striking similarities with the method in [1], which also seeks the optimization of a k-NN estimate of the state distribution entropy in a reward-free context. While it would be in general acceptable to overlook unpublished work, I think that the connections with [1] are too many to avoid a deeper discussion over distinctive contributions. The method in [1] does not seem to learn representations, but I am wondering if this contribution alone would be substantial enough.

C2) The paper does not present an explicit empirical evaluation of the reward-free phase, thus I am not sure on how APT is performing in entropy maximization. Especially, what is the impact of the three sources of bias that are introduced in the entropy estimation (avoiding bias correction and constants, avoiding importance weighting, scaling distances with the standard deviation)?
Can authors present state entropy plots, and possibly compare the performance of APT with other methods seeking a similar objective, such as MaxEnt (with representation learning), SMM [2] or MEPOL [1].

C3) I have some concerns on the theoretical and practical implications of considering a reward function that is actually depending on the current policy. It is sound to fit a value function for a reward of this kind? Maximizing an ever-changing reward might cause instability and prevent convergence?

C4) The benefit of employing a non-parametric method to estimate the entropy of high-dimensional inputs is clear, since density modeling would be quite hard. Could density modeling over a reduced latent space be a viable option instead?

C5) I would argue that the idea of simultaneously learning representations and exploration is the main selling point of the presented method, since maximizing the state entropy might help learning superior representations and viceversa. But from the ablation study in Section 4.3 this conclusion does not arise naturally, as learning representations alone seems almost as good as learning both. May I ask the authors to clarify this point?

C6) The scores on Atari games are reported without confidence intervals, leaving some doubts over the statistical significance of the results. Moreover, it is not completely clear from the aggregate performance where and how APT is helping in these experiments. Can authors provide a deeper explanation on why the original performance of SimPLe and VISR is not reproducible?

[1] Mutti and Restelli. A policy gradient method for task-agnostic exploration. Arxiv, 2020.
[2] Lee et al. Efficient exploration via state marginal matching. Arxiv, 2019

*QUESTIONS*

May the authors address the comments listed above in their response?

*ADDITIONAL FEEDBACK*
- For the Atari experiments I would suggest to focus more on hard-exploration games, such as Montezuma or Pitfalls, instead of providing just an aggregate performance over full sets of games. It would be nice to include some visualizations and interpretations on the behavior that APT learns in the reward-free phase.
- I believe that the multi-environment pre-training setting is quite interesting and, to the best of my knowledge, completely novel. The results are promising, and I would suggest to include this setting, together with a deeper analysis, in the main paper.
- Dashing the lines in the reported plots would help visibility, especially without colors.
- typos and rephrasing:
	- when referring to the environment I would use reward-free instead of task agnostic (e.g., page 2, paragraph 3)
	- the 26 games subset instead of the 100k subset (page 7, last lines)
	- I would rephrase "The notable difference is that APT (representation) decouples the action space dimension from pre-trained models" which is not crystal clear (Section 4.3).


####################

AFTER RESPONSE

I would like to thank authors for their detailed response and for their effort in improving the paper according to reviewers' suggestions. Unfortunately, after authors clarifications, I still have some doubts on the concerns raised with C1 and C2 (see below). Thus, I am keeping a slightly negative evaluation for this paper.

Authors claim that the main benefit of APT over a prior work method (MEPOL, Mutti er al., 2020) is a lower variance of the gradient estimation, thanks to the choice of avoiding importance weights corrections. However, in Figure 6 MEPOL does not seem to suffer a particularly high-variance. To me, the most likely reason for the improved performance is that APT guarantees an action-level feedback as opposed to a trajectory feedback (see [1]).
However, I am still skeptical about this action feedback: the reward-to-go becomes non-Markovian and Bellman equations does not hold anymore (see [2]). This casts some doubts on the actor-critic procedure APT employs to optimize the rewards. Authors may have a good point on the notion that the encoder is breaking the dependence between policy and rewards, but I think the topic warrant some additional discussion.

I would suggest the authors to rephrase this work to give a more central role to the scalability to high-dimensional observations, which I believe is the main contribution of the paper, and to include a more thorough discussion of (Mutti et al., 2020) in the main text (beyond the related work section).

[1] Efroni et al. Reinforcement learning with trajectory feedback. Arxiv, 2020.

[2] Zhang et al. Variational policy gradient method for reinforcement learning with general utilities. NeurIPS 2020.

---

> ### Author Response · Authors · 2020-11-20
> **Response to AnonReviewer3**
>
> Thank you for your feedback on our work.
>
>
> **Q**: **Comparison with similar method [1]** “Can authors present state entropy plots, and possibly compare the performance of APT with other methods seeking a similar objective, such as MaxEnt (with representation learning), SMM [2] or MEPOL [1].” “The paper does not present an explicit empirical evaluation of the reward-free phase....”
> **A**: Thank you for pointing out [1], it is very relevant to our work, we have included a discussion and empirical comparison in the revision. MEPOL is a concurrent work that also considers particle-based entropy maximization. They proposed the trust-region policy gradient optimization of entropy and showed significantly improved data efficiency over existing practices including SMM and MaxEnt.
> In terms of methodology, our work differs in that we propose to maximize the biased but lower-variance entropy estimation therefore our work gets rid of estimating high variance importance weights. Due to the simplicity of the estimation, our work can adapt state-of-the-art off-policy RL algorithms for higher data efficiency.
> In terms of results, MEPOL is only limited to state based RL while our work extends to visual domains and achieves state-of-the-art data efficiency in visually diverse RL benchmarks from Atari games to DMControl suite.
> We have conducted a thorough experiment to compare APT with MEPOL, the results are shown in Figure 6 and Table 3. We found that during the reward-free pre-training phase, APT achieves a significantly higher entropy index than MEPOL in high dimensional continuous control tasks. During the fine-tuning phase, APT outperforms MEPOL and ablated baselines and solves downstream continuous control tasks much more efficiently.
>
>
>
> **Q**: **How sound is it to fit value function to intrinsic rewards that change over time?** “I have some concerns on the theoretical and practical implications of considering a reward function that is actually depending on the current policy. It is sound to fit a value function for a reward of this kind? Maximizing an ever-changing reward might cause instability and prevent convergence?”
> **A**: Many practices are doing so, in fact, most intrinsic rewards depend on policy in various forms, e.g. curiosity (*Pathak et, al.*). Since these intrinsic rewards are computed within some latent space (contrastive representation space in our case), the reward is a stable signal to guide self-supervised RL. That being said, we agree that in general maximizing an ever-changing reward can cause instability if the representation learning part fails.
> Theoretically, we can prove that under certain assumptions of the MDPs, the intrinsic reward goes to zero as training total sample size goes to infinity, which is a favorable property for pre-training, we have included an analysis of the reward’s asymptotic behavior in the revision.
>
>
> **Q**: **Does count-based pre-training with density modeling over reduced latent space work well?** “The benefit of employing a non-parametric method to estimate the entropy of high-dimensional inputs is clear, since density modeling would be quite hard. Could density modeling over a reduced latent space be a viable option instead?”
> **A**: No, as shown in our ablation study. This ablated baseline is denoted as contrastive count-based pre-training, which trains a VAE on contrastive representation instead of raw pixel observations. We found it performs much worse than directly training from pixels with PixelRNN as density modeling.
>
>
> **Q**: **Fine-tuning representation and fine-tuning both representation and policy which one is better?** “I would argue that the idea of simultaneously learning representations and exploration is the main selling point of the presented method, … as learning representations alone seems almost as good as learning both. May I ask the authors to clarify this point?”
> **A**: On DeepMind control, we found fine-tuning representation performs almost as well as fine-tuning both. However, as given in Atari results, fine-tuning both significantly outperform fine-tuning representation only, especially on hard exploration tasks, indicating the importance of fine-tuning exploratory policy in visual diverse domains. We believe the discrepancy is due to Atari games having more complex environment dynamics than DMControl and requiring better exploration policy than DMControl suite.

---

> > ### Author Response · Authors · 2020-11-20
> > **Response to AnonReviewer3 (continue)**
> >
> > **Q**: **Include the behaviour and performance of the RL agent on Atari games during pre-training phase in the paper** “For the Atari experiments I would suggest to focus more on hard-exploration games, such as Montezuma or Pitfalls, instead of providing just an aggregate performance over full sets of games. It would be nice to include some visualizations and interpretations on the behavior that APT learns in the reward-free phase.”
> > **A**: Evaluating the agent during the entire pre-training phase is very computationally expensive, unfortunately, we do not have enough time or resources to complete within this author response window. We will include the results on a representative subset of the Atari games suite in the final version.
> >
> >
> > **Q**: **Highlight and include a deeper discussion regarding the multi-environment pre-training** “I believe that the multi-environment pre-training setting is quite interesting and, to the best of my knowledge, completely novel. The results are promising, and I would suggest to include this setting, together with a deeper analysis, in the main paper.”
> > **A**: We have revised the submission per your suggestions.
> >
> >
> > **Q**: **Edit suggestions** “The scores on Atari games are reported without confidence intervals, leaving some doubts over the statistical significance of the results... Can authors provide a deeper explanation on why the original performance of SimPLe and VISR is not reproducible?” “Dashing the lines in the reported plots would help visibility, especially without colors.”
> > **A**: The Atari results of our method are averaged over five random trials, we have included standard derivations in the revision. Since different papers report different values of SimPLe and other baselines, we choose the best available results to contrast with APT. We have improved visibility of the reported plots in the revision per your suggestions.
> >
> >
> >
> > [1] Mutti M, Pratissoli L, Restelli M. A Policy Gradient Method for Task-Agnostic Exploration. arXiv preprint arXiv:2007.04640.
> > [2] Lee L, Eysenbach B, Parisotto E, Xing E, Levine S, Salakhutdinov R. Efficient exploration via state marginal matching. arXiv preprint arXiv:1906.05274.
> > [3] Hazan E, Kakade S, Singh K, Van Soest A. Provably efficient maximum entropy exploration. International Conference on Machine Learning 2019 May 24 (pp. 2681-2691). PMLR.

---

### Official Review · AnonReviewer1 · 2020-10-26

**Rating:** 6
**Confidence:** 4

**Review:**

This submission presents a technique for unsupervised pre-training of representations and policies for RL. Unsupervised representation learning has obtained impressive results in supervised scenarios, and adapting these methods to RL is an important research direction. One of the main challenges in the RL setting is that of defining the distribution of data to learn from, as well as sampling from it. The learned representations are unlikely to be useful for observations that are out of the pre-training distribution, so it is desirable to perform representation learning on data that is representative of the full state space. Previous works (Hazan et al., Lee et al.) proposed strategies to train agents that induce maximally entropic state visitation distributions, but they involve density estimation whose underlying assumptions are not well suited for pixel observations. The authors propose to overcome these limitations by using a particle based entropy estimate in the learned representation space. The pre-trained representations and policies can be used for RL from pixels, obtaining faster convergence and higher end scores than the considered baselines in both DMControl and Atari.

The method is novel and the experimental results are strong. The paper includes a proper literature review and it is generally easy to follow. However, it would benefit from a more detailed analysis in order to understand where the reported gains come from. The ablation studies suggest that most of the gains are due to the learned representations: fine-tuning the policy as well only provides slightly faster convergence in two Hopper tasks, but the end performance is the same as when transferring representations only. I would like to suggest the following:
- Reporting the performance of zero-shot transfer for all experiments, i.e. the average return of the pre-trained policy in each task (APT@0). I suggest doing this for both DMControl and Atari experiments.
- It would be very helpful to see per-game scores in Atari in order to understand what type of environments benefit the most from pre-training. Gains could come from faster convergence on dense reward games thanks to the pre-trained representations, or from higher end performance on hard exploration tasks due to the exploratory behavior of the pre-trained policy. This is important given the fact that APT obtains much higher median human normalized scores than VISR while reaching much lower mean scores.
- Related to the previous point, comparing full APT and APT (representation) in a selection of Atari games requiring different degrees of exploration would provide insight on the type of challenges APT is addressing.
- APT (representation) can be understood as pre-training a state representation. How does this version of APT compare to RL from true state on DMControl? A similar analysis was reported by Srinivas et al. (Figure 7, see full reference below).

Please note that most of the suggestions above do not require running new experiments and can be addressed by providing additional information about the ones that are already reported in the paper.


Other comments:
- To the best of my knowledge, there doesn’t exist a standard procedure for fine-tuning policies, especially for actor-critic architectures such as the one in SAC. It would be helpful to include a description on how this is performed. Is the critic fine-tuned as well, or is it initialized from scratch?
- Section 3 mentions a robustness analysis showing the impact of varying k in kNN, but I didn’t see it in the paper. Is it referring to initial experiments, or to some ablation study that was not included for some reason?
- SimCLR fits representations using an objective that is based on cosine similarity, but the reward derived from the particle-based entropy estimate employs L2 distance. Is there a reason for this discrepancy?
- The subtitle “Supervised training @100k” in Table 1 is not very accurate, as two out of five rows use more than 100k interactions. I suggest using “Fully-supervised training” as the title and appending “@100k” to the name of the first three methods.
- A single unsupervised pre-training can be leveraged to solve multiple tasks as long as the environment does not change, as shown in the DMControl experiments. I believe this is one of the most appealing properties of the method and might not be highlighted enough in the paper.


The following works are relevant and might be worth citing:
- Laskin, Michael, Aravind Srinivas, and Pieter Abbeel. "Curl: Contrastive unsupervised representations for reinforcement learning". ICML 2020.
- Lee, Lisa, et al. "Efficient exploration via state marginal matching". arXiv preprint arXiv:1906.05274 (2019).

---

> ### Author Response · Authors · 2020-11-20
> **Response to AnonReviewer1**
>
> Thank you for your positive comments and feedback on our work.
>
> **Q**: **Compare APT, APT (representation) and VISR on Atari games with different degrees of hardness.** “it would benefit from a more detailed analysis in order to understand where the reported gains come from. The ablation studies suggest that most of the gains are due to the learned representations:...”  “It would be very helpful to see per-game scores in Atari … reaching much lower mean scores.” “Related to the previous point, comparing full APT and APT (representation) in a selection of Atari games ...”
> **A**: We have conducted experiments on Atari requiring different degrees of exploration to further compare APT and APT (representation), the results are shown in Section 4.3 and Table 2. We found both variants of APT achieve significantly higher end performance than VISR and from scratch in hard exploration games. APT significantly outperforms APT(representation) in hard exploration while maintaining very high scores in the reset of games, indicating the exploratory policy in APT is particularly useful for solving difficult Atari games. As we observed in DMControl experiments, APT tends to have significantly higher zero-shot scores than APT (representation) in tasks that are extremely difficult for training from scratch.
>
>
> **Q**: **How does APT compare with training from scratch with states as input?**  “APT (representation) can be understood as pre-training a state representation. How does this version of APT compare to RL from true state on DMControl? A similar analysis was reported by Srinivas et al. (Figure 7, see full reference below).”
> **A**: We have included the results of training from scratch with states as input for comparison, shown in Appendix D. APT performs on par or better than state-based data-efﬁciency on most of the easier dense reward environments, but archives significantly better data-efficiency on the rest hard environments.
>
>
> **Q**: **Is k chosen empirically during initial experiments?** “Section 3 mentions a robustness analysis showing the impact of varying k in kNN, but I didn’t see it in the paper. Is it referring to initial experiments, or to some ablation study that was not included for some reason?”
> **A**: Yes, we fixed k=3 in all our experiments because we found it works better than larger values during initial experiments. If you would like, we will include an ablation study of k in the final version.
>
>
> **Q**: **Why using cosine similarity for contrastive learning but L2 distance to maximize the entropy** “SimCLR fits representations using an objective that is based on cosine similarity, but the reward derived from the particle-based entropy estimate employs L2 distance. Is there a reason for this discrepancy?”
> **A**: For SimCLR loss, the normalization and projection network are important components for learning good representations, therefore cosine similarity is used to measure distance, while for entropy maximization, the particle-based entropy is computed in the representation space therefore L2 distance is used.
>
> **Q**: **Does fine-tuning consist of tuning both actor and critic or only critic?** “To the best of my knowledge, there doesn’t exist a standard procedure for fine-tuning policies, especially for actor-critic architectures such as the one in SAC. It would be helpful to include a description on how this is performed. Is the critic fine-tuned as well, or is it initialized from scratch?”
> **A**: In the fine-tuning phase, APT means both actor and critic are fine-tuned, while APT(representation) means the encoder is fine-tuned but the actor and critic fully connected layers are randomly initialized. We have included a more clear description in the revision.
>
> **Q**: **Highlight APT is capable of doing multi-environment pre-training** “A single unsupervised pre-training can be leveraged to solve multiple tasks as long as the environment does not change, as shown in the DMControl experiments. I believe this is one of the most appealing properties of the method and might not be highlighted enough in the paper.”
> **A**: We have highlighted the multi-environment pre-training setting and results in the revision.
>
>
> **Q**: **Edit suggestions** “The subtitle “Supervised training @100k” in Table 1 is not very accurate, as two out of five rows use more than 100k interactions. I suggest using “Fully-supervised training” as the title and appending “@100k” to the name of the first three methods.” “more references”
> **A**: We have made changes per your suggestions in the revision.

---

> > ### Comment · AnonReviewer1 · 2020-11-23
> > **APT@0 and subsets of games**
> >
> > Thanks for addressing some of the concerns and questions in my review. I believe that there are two aspects that were not fully addressed in the reply that would help understanding where the gains provided by APT come from:
> >
> > - **APT@0:**: would it be possible to include scores for APT after the unsupervised stage to understand how much of the gain comes from the exploratory behavior of the pre-trained policy? Please note that this does not require running the reward-free training phase again.
> >
> > - **Scores per game:** I appreciate that authors included results for individual games. Would it be possible to include a table with scores for each of the 57 Atari games? The selected "hard exploration" games might not be the most representative ones, and I wonder what the performance of APT in games like Montezuma's Revenge or Pitfall is. I would recommend reporting scores before and after fine-tuning (i.e. APT@0 and APT@100k).

---

> > > ### Author Response · Authors · 2020-11-25
> > > **response to AnonReviewer1**
> > >
> > > Thank you again for your feedback.
> > >
> > > **Q**: **How does APT work in zero-shot settings?** **include the results of APT and zero-shot performance on each Atari game** "would it be possible to include scores for APT after the unsupervised stage..." "Would it be possible to include a table with scores for each of the 57 Atari games? ... I would recommend reporting scores before and after fine-tuning (i.e. APT@0 and APT@100k)."
> > > **A**: We have included the breakdown results of APT and APT$@0$ on each Atari game in the revision. The results are shown in Appendix E and Section 4.3. We found that the APT$@0$ is outperformed by APT$@100k$ and VISR$@100k$ in terms of median and mean scores, in some exploration games, the zero-shot scores of APT are higher than VISR$@100k$, showing the benefit of pre-trained exploratory behaviour. APT$@100k$ significantly outperforms APT$@100k$ on hard exploration games, especially sparse reward exploration games, but performs on par or worse than VISR$@100k$ on the rest of dense reward games. We believe the reason behind this is that the exploratory pre-trained model in APT is extremely useful for hard exploration games while VISR can adapt to dense reward functions quicker than APT due to its successor feature nature.

---

### Official Review · AnonReviewer2 · 2020-10-28
**important direction, simple scheme, but some concerns about experiments and novelty**

**Rating:** 5
**Confidence:** 3

**Review:**

The paper presents a pre-training scheme (APT) for RL with two components: contrastive representation learning and particle based entropy maximization. Experiments are done in DeepMind Control Suite (DMControl) and Atari suite to show improved performance and data efficiency.

**Strength**: Pre-training good representations and policy initialization without reward is obviously an important direction in RL. The proposed method is conceptually simple and intuitive, yet achieves some promising results. Particle based entropy estimation (eq. (5)) can be a simple and interesting solution to estimating observation entropy in high dimension space.

**Weakness**: Some concerns about experiment results:
- For DMControl results, it seems APT only improves significantly over "From scratch" and "Count-based pre-training" schemes when reward is sparse? I also think the result can be made much stronger if more powerful baselines can be adopted, e.g. DIAYN, VISR, etc.
- For Atari results, APT versus VISR is interesting. APT achieves better median normalized scores but VISR achieves better average normalized scores. Is it possible to show a game breakdown to understand where these methods work better?
- In section 4.3, it is awkward that I find APT and APT (representation) similar across different curves in Figure 4, so I'm not convinced the policy initialization is useful. Maybe some more results (e.g. on Atari) can help with this.
Also see the **Novelty** part below.

**Novelty**: The two cores of the paper, contrastive representation learning and entropy maximization, are quite established in RL. Particle based entropy estimation is from prior work, but I think it is fairly novel and interesting in the RL domain (despite its usefulness being doubted).

**Clarity**: I think the paper is written clearly in general.

**Question**: Conceptually eq. (5) looks quite noisy (dependent on batch samples a lot), and like some form of contrastive learning objective still (maybe entropy maximization is equivalent to making representation contrastive?). So is it really a stable reward signal? Is it possible to just use eq. (1) as reward also?

---

> ### Author Response · Authors · 2020-11-20
> **Response to AnonReviewer2**
>
> Thank you for your feedback on our work.
>
> **Q**: **Does APT only improve significantly over “From scratch” and “count-based pre-training” schemes when reward is sparse?** “For DMControl results, it seems APT only improves significantly over "From scratch" and "Count-based pre-training" schemes when reward is sparse?”
> **A**: Yes per our evaluations on DMControl. The reason might be the exploration in DMControl dense reward tasks is not challenging therefore the pre-trained exploratory policy in APT is not significantly useful. Additionally, we have reported results on Atari games with different levels of exploration hardness. As shown in Table 2, we found APT significantly outperforms all baselines in hard exploration while maintaining very high scores in the reset of games. Further improving APT on the dense reward environments is an interesting direction.
>
>
> **Q**: **For DMControl, how does APT compare with VISR and DIAYN?** “I also think the result can be made much stronger if more powerful baselines can be adopted, e.g. DIAYN, VISR, etc.”
> **A**: We have evaluated VISR and DIAYN on DMControl, the complete results are given in Table 7 (Appendix). We found DIAYN has mixed results compared with training from scratch, VISR performs best in dense reward tasks and outperforms training from scratch in multiple tasks, and APT outperforms VISR by a large margin in most tasks.
>
>
> **Q**: **Explaining why APT achieves higher median but lower mean human normalized scores compared with VISR?** “For Atari results, APT versus VISR is interesting. APT achieves better median normalized scores but VISR achieves... where these methods work better?”
> **A**: APT performs significantly better than VISR in hard exploration games, e.g. Freeway, but VISR performs better in some dense reward easy exploration games, e.g. BattleZone. Therefore APT has higher median but lower mean human normalized scores compared with VISR. The reason might be APT focuses on exploration while VISR focuses on fast adaptation. We remark that it is straightforward to combine APT and VISR to have the best of both.
>
> **Q**: **Comparing APT and APT (representation) on Atari** “In section 4.3, it is awkward that I find APT and APT (representation) similar across different curves in Figure 4, so I'm not convinced the policy initialization is useful. Maybe some more results (e.g. on Atari) can help with this.”
> **A**: We have conducted experiments on Atari games to compare APT and APT (representation). In contrast to DMControl results, as shown in Table 2, APT performs significantly better than APT (representation) in Atari games, especially hard exploration sparse reward games. We believe the discrepancy is due to Atari games having more complex environment dynamics than DMControl and requiring better exploration policy than DMControl suite.
>
>
> **Q**: **What is the connection between contrastive learning and entropy maximization? Is it possible to use contrastive learning errors as reward?**  "Conceptually eq. (5) looks like some form of contrastive learning objective still (maybe entropy maximization is equivalent to making representation contrastive?). Is it possible to just use eq. (1) as reward also?"
> **A**: If the projection head in SimCLR is identity and there is no normalization after the projection head, then the denominator of SimCLR loss is ‘nearly’ the same as our particle-based entropy maximization. The difference is entropy maximization only pushes apart from its kth nearest neighbor while SimCLR pushes apart from all samples. Despite the difference, we agree that contrastive learning and entropy maximization can be unified together with certain modifications. In practice, we found the MLP projection head in SimCLR is important though, we leave exploring this interesting point as future work.

---

> > ### Author Response · Authors · 2020-11-20
> > **Response to AnonReviewer2 (continue)**
> >
> > **Q**: **Is the intrinsic reward noisy?** "Conceptually eq. (5) looks quite noisy (dependent on batch samples a lot), so is it really a stable reward signal?”
> > **A**: We have found using a large batch size is good enough to train the RL agent. That being said, we agree the noise in the reward definitely slowed down the pre-training. But ideally we should use the entire replay buffer or a huge batch size, in the revision, we have included more discussion on the reward, as shown in Lemma 1. Since we have limited computation resources, we made the choice of using the current batch to compute the reward. We remark that most current pre-training successes are computationally expensive., our work indicates more promising results if using larger batch size.
> >
> >
> > **Q**: **Novelty of combining contrastive learning and particle-based entropy maximization** “The two cores of the paper, contrastive representation learning and entropy maximization, are quite established in RL. Particle based entropy estimation is from prior work, but I think it is fairly novel and interesting in the RL domain (despite its usefulness being doubted).”
> > **A**: Previous studies of contrastive learning in RL are in a fully supervised setting, either with dense reward [1] or expert demonstrations [2], while our work focuses on unsupervised learning. Previous studies of entropy maximization in RL mostly concerned policy entropy maximization, studies of state entropy maximization require density models, making them fail in visual diverse domains like Atari and DeepMind Control [3,4].
> > To the best of our knowledge, for DeepMind Control, APT is the first one to successfully learn pre-trained models that can solve different downstream tasks as long as the environment does not change, and APT can do multi-environment pre-training, for Atari games, our work buys the performance of fully supervised learning with a fraction of total samples and performs on par or better than other pre-training methods. We believe being simple does not impair the method’s contributions to the community, our careful implementation choices and extensive experiments will allow wide adoption of this method.
> >
> >
> >
> > [1] Srinivas A, Laskin M, Abbeel P. Curl: Contrastive unsupervised representations for reinforcement learning. International Conference on Machine Learning 2020.
> > [2] Stooke A, Lee K, Abbeel P, Laskin M. Decoupling representation learning from reinforcement learning. arXiv preprint arXiv:2009.08319.
> > [3] Lee L, Eysenbach B, Parisotto E, Xing E, Levine S, Salakhutdinov R. Efficient exploration via state marginal matching. arXiv preprint arXiv:1906.05274.
> > [4] Hazan E, Kakade S, Singh K, Van Soest A. Provably efficient maximum entropy exploration. International Conference on Machine Learning 2019 May 24 (pp. 2681-2691). PMLR.

---

### Decision · Program_Chairs · 2021-01-07
**Final Decision**

**Decision:**

Reject

**Comment:**

The authors propose a particle-based entropy estimate for intrinsic motivation for pre-training an RL agent to then perform in an environment with rewards. As the reviewers discussed, and also mentioned in their reviews, this paper bears stark similarity to work of 5 months ago, presented at the ICML 2020 Lifelong ML workshop, namely, "A Policy Gradient Method for Task-Agnostic Exploration", Mutti et al, 2020--MEPOL. What is novel here is the adaptation of this entropy estimate to form an intrinsic reward via a contrastive representation and the subsequent demonstration on standardized RL environments.  The authors have added a comparison to MEPOL, and in these experiments, APT outperforms this method, sometimes by some margin. Unfortunately this work does not meet the bar for acceptance relative to other submissions.